# Application of Additive Manufacturing and Deep Learning in Exercise State Discrimination

**DOI:** 10.3390/s25020389

**Published:** 2025-01-10

**Authors:** Zhilong Zhao, Jiaxi Yang, Jiahao Liu, Shijie Soong, Yiming Wang, Juan Zhang

**Affiliations:** 1Biomanufacturing Center, Department of Mechanical Engineering, Tsinghua University, Beijing 100084, China; zhaozhilong@mail.tsinghua.edu.cn (Z.Z.); yiming_wang@tsinghua.edu.cn (Y.W.); 2USC Viterbi School of Engineering, Daniel J. Epstein Department of Industrial and Systems Engineering, University of Southern California, Los Angles, CA 90089, USA; jyang168@ucsc.edu; 3School of Instrument Science and Opto-Electronics Engineering, Hefei University of Technology, Hefei 230001, China; 2022215153@mail.hfut.edu.cn; 4Department of Bioengineering, Royal School of Mines, Imperial College London, London SW7 2AZ, UK; shijie.academic@gmail.com

**Keywords:** bioelectrical signal, reverse engineering, additive manufacturing, long short-term memory (LSTM) neural network, sports fatigue states

## Abstract

With the rapid development of sports technology, smart wearable devices play a crucial role in athletic training and health management. Sports fatigue is a key factor affecting athletic performance. Using smart wearable devices to detect the onset of fatigue can optimize training, prevent excessive fatigue and resultant injury, and increase efficiency and safety. However, current wearable sensing devices are often uncomfortable and imprecise. Furthermore, stable methods for fatigue detection are not yet established. To address these challenges, this paper introduces 3D printing and deep learning to design a smart wearable sensing device to detect different states of sports fatigue. First, to meet the need for comfort and improved accuracy in data collection, we utilized reverse engineering and additive manufacturing technologies. Second, we designed a prototype based on the long short-term memory (LSTM) neural network to analyze the collected bioelectrical signals for the identification of sports fatigue states and the extraction of related indicators. Finally, we conducted a large number of numerical experiments. The results demonstrated that our prototype and related equipment could collect signals and mine information as well as identify indicators associated with sports fatigue in the signals, thereby improving accuracy in the classification of fatigue states.

## 1. Introduction

In the fields of deep-sea operations, sports, and aerospace, the physical and mental states of personnel are often under tremendous pressure, and real-time physical-state monitoring must be performed in order to avoid casualties and property loss from excessive fatigue [1]. Therefore, it is necessary to identify the contemporaneous fatigue status of those engaged in the above-noted fields. Consequently, monitoring fatigue status conveniently, cheaply, quickly, and noninvasively has become a concerted demand in many fields [2]. Among them, in professional sports, competitors are subjected to exhaustive training, along with the resultant stresses, both physical and psychological [3]. Chronic stress can cause exercise fatigue, leading to exercise burnout, compromised immunity, and an increased risk of disease and infection [4,5]. As exercise progresses, fatigue increases, along with changes in the physiological electrogram signal. This explains why researchers have studied changes in electrical signals during exercise to extract signal indicators related to exercise fatigue [6].

In recent years, methods for extracting physiological electrical signals for fatigue detection have developed rapidly. Sheykhivand et al., for instance, designed a deep convolutional neural network–long short-time memory (CNN-LSTM) network to extract characteristics from raw EEG data corresponding to six active areas, denoted as A, B, C, D, and E, based on a single channel, and F, for automatic fatigue detection systems [7]. Chattopadhyay et al. also presented a framework to identify fatigue states. Specifically, the proposed framework acquires time-series data from a surface electromyogram (sEMG) sensor and employs state-of-the-art machine learning methods to measure physiological status [8]. Moreover, Feng et al. used specialized electrocardiogram (ECG) acquisition devices to collect ECG data during training sessions. Subsequently, selected ECG indicators with statistical significance, particle swarm optimization–support vector regression (PSO-SVR), and extreme learning machine (ELM) were used to predict physical fatigue [9]. However, noninvasive, mature, stable, and commercially available monitoring solutions on a large scale have yet to emerge.

The human body is a complex organism, and the recognition of the fatigue state involves many complex factors; therefore, a single index cannot be used to effectively detect the fatigue state. Past methods have used bioelectric signal pattern recognition. These traditional pattern recognition methods included support vector machines (SVMs) and artificial neural networks (ANNs) [10]. However, traditional pattern recognition classification is only as good as the input feature signal, and such feature selection can be confounded by subjectivity and uncertainty. This necessitates carrying out complex preprocessing of the signal, followed by choosing the appropriate method for feature extraction, feature selection, fusion, and classification. Such methods can be easily derailed by such disadvantages as the high number of parameters, complicated calculations, and long operation time.

The recognition of the fatigue state involves multiple complex physiological and psychological factors. Traditional methods relying solely on bioelectric signal pattern recognition face inherent limitations in capturing the comprehensive nature of sports fatigue. Current research in sports science has demonstrated that exercise fatigue manifests itself through a complex interplay of various physiological systems. Mental stress impacts cognitive function and reaction time, while sleep quality directly affects recovery and performance capacity. Additionally, nutritional status influences energy availability and metabolic responses during exercise. Environmental conditions, including temperature, humidity, and altitude, can significantly alter physiological responses and accelerate fatigue onset. These factors collectively contribute to the overall fatigue state, suggesting that a more integrated approach to fatigue assessment might be necessary. While bioelectric signals provide valuable objective measurements, their interpretation should be contextualized within this broader physiological framework. This complexity necessitates the development of more comprehensive monitoring systems that can account for multiple physiological and environmental indicators to provide a more robust assessment of fatigue states.

Therefore, we envisioned employing a solution for detecting the athletic fatigue state by combining multiple physiological indicators with deep learning methods to support a bio-surface electrical signal extraction device based on additive manufacturing (Figure 1). Additive manufacturing technology is becoming increasingly popular in the field of customized equipment owing to its short production cycle and low manufacturing costs. For devices that require high precision and a high degree of customization, the time from design to production can be substantially reduced by using 3D-printing methods. This allows for the creation of unique customized devices to meet the specific needs of individuals [11]. These characteristics coincide with the needs of professional athletes who have to participate in a wide range of events around the world and prepare for competitions in multiple regions. Meanwhile, additive manufacturing technology can be a sound solution to the problems of wearability and testing accuracy that exist widely in commercially available sensing devices [12]. Moreover, compared with manual feature extraction using traditional pattern recognition methods, deep learning methods can automatically identify relevant features in data, and these methods are more suitable for handling complex, unstructured data, such as images, audio, and text. As data volume increases, deep learning models often improve in performance, whereas traditional models might plateau. The ability of deep learning methods to learn hierarchical feature representations is particularly advantageous for classifying physiological signals.

## 2. Reverse-Engineering-Based Additive Manufacturing for Wearable Device Design

In situations involving the long-term outdoor use of wearable devices, the design criteria extend beyond just accuracy and sensitivity [13]. For athletes, key aspects of wearability include comfort, portability, and anti-slip features, which also contribute to improving the quality of data collection. Accordingly, the design of these sensing devices prioritizes a compact and lightweight construction to better suit the active and outdoor nature of their use. This approach ensures that wearables meet the specific needs of athletes while maintaining high functionality.

In this paper, we use reverse engineering and additive manufacturing techniques to customize wearable sensing devices for a target population, resulting in significant improvements in wearability and signal collection efficiency. Our goal is to create a superior wearable device that integrates precise signal analysis and monitoring, taking into account user comfort and robust structural integrity. This method involves personalized modeling and sampling facilitated by reverse engineering, while additive manufacturing streamlines the production process and curtails costs.

Device design is preceded by a scan of the user’s intended wearable area, creating a 3D computer model for precise device tailoring. As shown in Figure 2, the outer shell of the device is shaped to the contours of the muscles, optimizing comfort by reducing skin pressure and enhancing grip. This design maintains a compact size, while incorporating essential components. The final product, a result of quick 3D printing and assembly, offers a refined balance between minimalism and functionality in a wearable form.

The transition from prototype to mass production involves several key considerations. The current manufacturing process utilizes industrial-grade 3D printers with a printing precision of ±0.1 mm, capable of producing up to 100 units per day per printer. The primary material selected is medical-grade TPU (thermoplastic polyurethane), which offers excellent biocompatibility and durability while maintaining cost-effectiveness at scale. The total component cost can be reduced by approximately 45% through bulk purchasing and optimized manufacturing processes compared with prototype costs. Quality control measures include automated dimensional verification, standardized electrode-positioning tests, and waterproof testing for each unit. The modular design allows for the automated assembly of 85% of the components, significantly reducing labor costs and production time. Initial pilot production runs demonstrate a yield rate of 90%, with most defects identifiable during the automated testing phase.

The versatility of 3D printing in the design phase enables the incorporation of artistic elements or complex structures into the device shell, which is then tailored to meet individual preferences. Following these steps, the prototype undergoes user testing, in which feedback drives specific enhancements for a more refined experience. This iterative process ensures that the device aligns with the user’s requirements, thereby enhancing personalization and performance. Feedback and improvements are methodically archived, serving as valuable references for future sensor development.

## 3. Analysis of Signal Characteristics of the Athletic Fatigue State Based on Surface EMG

While surface EMG and ECG signals offer considerable insights into exercise fatigue assessment, the interpretation and application of these bioelectrical signals must be considered within a broader physiological context. Exercise fatigue manifests through complex interactions between neuromuscular, cardiovascular, and metabolic systems, which are influenced by both internal physiological states and external environmental factors. The psychological state of athletes significantly impacts their fatigue perception and performance capacity. Sleep patterns affect recovery mechanisms and subsequent exercise performance, while nutritional status influences the metabolic efficiency and energy availability. Environmental conditions such as ambient temperature, humidity, and altitude can alter physiological responses and affect the signal acquisition quality. Our analysis acknowledges these multifaceted influences while focusing on the specific contributions of bioelectrical signals to fatigue assessment, establishing the groundwork for future comprehensive monitoring systems.

### 3.1. Preprocessing of Surface EMG Signal Filtering

The goal of processing dynamic body-surface bioelectric signals is to discern vital fatigue indicators from EMG and ECG signals collected during physical activity. Initially, filtering the original signal is necessary to remove diverse types of noise. The process involves analyzing the relationship between extracted features and sports-driven fatigue and then selecting indicators with significant feature trends for constructing the fatigue model. The quality of initial signal processing critically influences the accuracy of feature extraction and the scientific validity of fatigue assessment [14]. Based on the hardware intended to detect physiological changes in response to sports-specific fatigue, the surface EMG signal energy predominantly lies in the 20 to 150 Hz range, with EMG signal interference that arises from such specific frequency range disruptions as baseline drift, power frequency interference, and motion artifacts [15].

For surface EMG signals, a high-pass filter with a 20 Hz cutoff frequency was applied to eliminate baseline drift while preserving energy in the critical 20–150 Hz range, and additional filtering addressed 50 Hz power interference and 3–14 Hz motion artifacts. For ECG signals, a fast median filter was used to remove baseline drift while retaining key features like R-wave peaks for accurate heart rate (HR) and heart rate variability (HRV) calculations. These preprocessing steps effectively remove noise, maintain signal integrity, and ensure that the extracted features reliably reflect physiological fatigue changes.

(1)Baseline drift

Baseline drift, a type of low-frequency noise in bioelectric signal acquisition, is influenced by factors like system changes and human respiration. This drift is a low-frequency signal typically ranging from 0.15 Hz to a few Hz. To filter out this drift, a high-pass filter with a 20 Hz cutoff frequency is first calculated by fc = wc/2π = 1/(2π) RC, where fc represents the cutoff frequency, wc is the angular frequency, R is the resistance, and C is the capacitance. The results can be effectively utilized in designing the hardware’s circuitry, ensuring cleaner signal acquisition and more accurate data analysis.

(2)Power frequency interference

The human body, acting as a conductor, is susceptible to electromagnetic radiation, while alternating current runs at 50 Hz power frequency interference, leading to significant common-mode interference. This interference, centered around a fluctuating 50 Hz frequency that results from the variable electromagnetic environment, can then overshadow the small amplitude of surface EMG signals. To mitigate this, we can implement hardware trap circuits and other methods specifically designed to filter out these interferences, ensuring a clearer capture of the surface EMG signal.

(3)Motion artifacts

During physical activity, limb movements can cause motion artifacts in signal acquisition. These artifacts arise as the electrode sheet is stretched or compressed, altering its position relative to the skin, thus changing the electrical properties and bioelectric potential of the skin–electrode interface. This introduces noise to the useful signal. Given that motion artifacts primarily occur in the 3–14 Hz frequency range, designing a classic hardware filter circuit with a 20 Hz high-pass cutoff frequency effectively eliminates these interferences, enhancing the signal’s clarity and reliability.

Figure 3 presents a comparative analysis of EMG signal waveforms pre- and post-filtering. Figure 3a displays the raw surface EMG signal, including the motion artifact and baseline drift interference, as being characterized by random and large amplitudes. Figure 3b shows the EMG signal post-20 Hz high-pass filtering, effectively isolating the EMG waveform by filtering out motion interference and baseline drift. Figure 3c,d depict the surface EMG signal waveforms before and after power frequency filtering, demonstrating the impact of this filtering process.

### 3.2. Feature Extraction of Surface EMG Signals as a Reflection of Exercise Fatigue

To analyze the correlation between EMG signals and exercise fatigue, EMG data were examined across three different levels of fatigue, corresponding to various scores on the exercise fatigue awareness scale. The EMG signals of two test muscles were studied at rate of perceived exertion (RPE) values of 8, 13, and 18, representing states of being very relaxed, slightly tired, and very tired, respectively. Figure 3 illustrates these surface EMG signals, with the top, middle, and bottom parts showing 10 s continuous EMG recordings at RPE values of 8, 13, and 18.

The results in Figure 4 reveal distinct patterns in surface EMG signals correlating with different fatigue states. Initially, the EMG signal shows regularity, high frequency, and low amplitude, aligning with the subject being in a relaxed state. As fatigue progresses slightly, the frequency remains about the same, but the amplitude increases. In a state of high fatigue, however, the frequency diminishes, and the amplitude rises significantly. Thus, this trend indicates that with increasing exercise fatigue, the EMG signal frequency tends to decrease while its amplitude progressively increases.

Our analysis focuses on surface EMG signals detected in various states of exercise fatigue. It incorporates both time-domain and frequency-domain indicators from the EMG signal. Time-domain analysis indicators include the root mean square (*RMS*), average rectified value (*ARV*), and integrated EMG value (*IEMG*) [16]. These indicators are crucial for quantifying muscle activation and fatigue, as calculated by the following specific equations, which accurately evaluate the signal characteristics related to exercise fatigue.(1)$$ARV=\frac{1}{T}\int_{t}^{t+T}\left|EMG(t)\right|dt$$(2)$$IEMG=\int_{t}^{t+T}\left|EMG(t)\right|dt$$(3)$$RMS=\sqrt{\frac{1}{N}\int_{t}^{t+T}EMG{\left(t\right)}^{2}dt}$$

In the frequency-domain analysis of surface EMG signals, the time-domain EMG signal undergoes Fast Fourier Transform (FFT) to obtain the power spectrum and spectral change data. This approach helps to elucidate characteristics of the EMG signal in the frequency dimension. Key indicators, like the mean power frequency (*MPF*) and the median frequency (*MF*), are used to define these features [17]. These metrics provide a deeper understanding of the EMG signal’s frequency-based properties, again with specific equations established for precise analysis.(4)$$MPF=\frac{\int_{0}^{\infty }f\cdot PSD(f)df}{\int_{0}^{\infty }PSD(f)df}$$(5)$$MF=\frac{1}{2}\int_{0}^{\infty }PSD(f)df$$

In the exercise fatigue test, surface EMG signals were collected from two leg muscles: lateral femoris, rectus femoris, and medial femoris. After filtering the original signals, both time-domain and frequency-domain characteristics were extracted, including the ARV, IEMG, RMS, MPF, and MF. These indicators facilitated a detailed sports fatigue correlation analysis. Table 1 presents EMG indicator outcomes, with the data normalized to <100 for visual clarity. The histogram in Figure 5, which is based on the Table 1 data, reveals variations in the above indicators across different exercise fatigue levels, showing a progressive increase in the time-domain indexes (RMS, ARV, and IEMG) and featured index values along the gradient of exercise.

The RMS, with its significant variability, serves as a key time-domain index for muscular fatigue. As exercise fatigue increases, the frequency-domain markers MPF and MF tend to decrease, with the MPF showing greater variation and stability. Thus, MPF is used as a primary frequency-domain index in assessing exercise fatigue [18]. The analysis of EMG signals from the two leg muscles demonstrates a consistent trend based on changes in the values of characteristics arising from different fatigue levels. This consistency indicates that the signals provide an objective reflection of the physiological response, i.e., fatigue. However, notable differences in time- and frequency-domain markers, especially in the lateral femoral muscle, still present challenges in precise fatigue measurement [19]. Therefore, in this study, MPF (frequency-domain) and RMS (time-domain) are selected for the further classification of exercise fatigue.

## 4. Analysis of the Characteristics of Exercise Fatigue Based on the ECG Signal

### 4.1. Preprocessing ECG Signal Baseline Drift

The ECG signal, a weak bioelectric signal from the body’s surface, primarily resides in the 0.05–100 Hz frequency range, with key components between 0.5 and 20 Hz. Baseline drift, overlapping with the low-frequency spectrum of the ECG, requires software filtering algorithms to remove unwanted noise [20]. Analysis shows that the fast median-filter algorithm, a nonlinear signal processing method based on order statistics theory, excels in handling ECG signal baseline drift. Established in 1970, median filtering is effective in impulse noise reduction and preserving signal edges, making it the preferred choice in signal noise suppression.

The principle of median filtering is as follows [21]: We assume that the ECG signal is x(n)(n=1,2,…,N); then, we take the length of the filter window as L=2K+1 (K is a positive integer) and input the signal of the window at time n. Then, the sequence sampling points are x(n−K),…,x(n) ,…x(n+K). The output at this time is:(6)y(n)=Mid{x(n−K),…,x(n),…x(n+K)}
where $Mid\left\{\right\}$ operation means that a sorting algorithm is adopted to arrange all the values in the filtering window from small to large, effectively removing values in the middle.

Inferring from Equation (6), it can be seen that sorting data in the filter window from small to large is the main calculation amount, so the key to determining the calculation speed of the median-filter algorithm is the calculation speed of the data-sorting algorithm in the filter window. We assume that the length of the filter window is set to 2K+1, expressed as (m(1),m(2),…,m(2K+1)); that the original data of the input ECG signal to be processed are N, expressed as (x(1),x(2),…,x(N)); and that the median-filter algorithm calculation times are (N−2K)∗(2K+1) times. According to theoretical algorithm complexity estimation, when the input data are n, the minimum algorithm complexity is $O(n\log_{2}n)$. Therefore, the calculation time and resource consumption of the traditional median-filter algorithm are determined by the amount of data to be processed and the size of the filter window.

According to the characteristics and limitations of the traditional median-filter algorithm, the fast median-filter algorithm, which optimizes the traditional median-filter algorithm, significantly improves the calculation speed of the filter algorithm. The fast median-filter algorithm first sorts the original data (x(i),x(i+1),…,x(i+2K)) in the window, then arranges the original data from small to large, and finally determines the number of sequences constituting the median value. When the first sorting is completed, (m(i),m(i+1),…,m(i+2K)) is the sorted ordered sequence. Then, the algorithm performs a second sorting by slightly moving the filter window down to obtain the new original data sequence, (x(i),x(i+1),…,x(i+2K)). Then, the first sorted sequence, (m(i),m(i+1),…,m(i+2K)), is sorted with the newly added data x(i+2K+1). The second sorting only needs internal interpolation and a binary search to add x(i+2K+1) as an ordered sequence. To avoid the reordering of n data processes, the algorithm reduces the number of operations and improves the efficiency of the filtering algorithm. The fast median filtering of the original ECG signal is shown in Figure 6.

### 4.2. Extracting the Exercise Fatigue Feature of ECG Signals

To analyze the link between ECG signals and exercise fatigue, ECGs corresponding to the three fatigue levels on the exercise fatigue awareness scale were selected. This selection aims to identify trends in the various ECG signal indicators as exercise fatigue deepens. Figure 7 illustrates these trends using RPE values of 8, 13, and 18, corresponding to 10 s ECG data captured under different fatigue states: very relaxed, slightly tired, and very tired. This approach allows for a nuanced understanding of how fatigue levels are reflected in ECG readings.

According to the ECG signal analysis provided in Figure 6, when the subject feels very relaxed, the ECG appears smoother and more regular with a wider RR gap. As the subject begins to feel slightly tired, the RR gap narrows, but the ECG signal amplitude changes only a little. However, when the subject feels very tired, both the ECG amplitude and RR gap are reduced, along with a noticeable increase in heart rate. ECG signals are rich in characteristics crucial for understanding physiological states. For example, a regular-rhythm ECG has regular P waves preceding a QRS complex in a regular rhythm. Therefore, QRS complex detection, particularly the R wave peak, is key for calculating heart rate and variability using RR intervals. In the study of Tuncer et al. (2020), for example, the subject’s ECG signal was processed using MATLAB to filter out interferences, and an adaptive threshold detection algorithm was used for R wave detection and HRV data acquisition [22].

We analyzed ECG indicators, including heart rate (HR), high-frequency power (HF, 0.15–0.4 Hz), low-frequency power (LF, 0.04–0.15 Hz), and the LF/HF ratio, under different states of athletic fatigue. The results, presented in Table 2 as “mean ± standard deviation”, are visually depicted in Figure 8. The analysis shows clear differences among HR, LF, and HF values across varying fatigue states, whereas the LF/HF variation is minimal, making it less suitable for fatigue detection. Based on these findings, HR, LF, and HF were selected as inputs for fatigue detection analysis.

## 5. Real-Time Monitoring of Sports Status Is Accordant with the LSTM-Based Recurrent Neural Network (RNN) Model

### 5.1. LSTM-Based RNN

The hidden layer structure of the RNN model enables the network to retain output information, thereby using both current and previous outputs to determine the current outcomes. RNNs have improved traditional neural networks by featuring interconnected hidden-layer nodes. Moreover, the addition of long short-term memory (LSTM) to RNNs addresses issues of gradient explosion and disappearance in long-distance information transfer, ensuring the sustained influence and relevance of historical sequence data. This innovation in the LSTM architecture effectively maintains information integrity over time.

The standard RNN hidden-layer unit contains only one tanh (hyperbolic tangent) layer, the structure of which is elementary. The RNN repeating-unit structure is shown in Figure 9a, where $X_{t}$ represents the input, $H_{t}$ represents the output, and A represents the hidden layer. The LSTM network essentially improves upon the standard RNN by the addition of a hidden-layer modular structure, as shown in Figure 9b.

The LSTM neural network unit controls the discarding or adding of information from the cell state through the structure of some “gates”. Specifically, the “forget gate” is used to control the degree to which the information in the cell state should be discarded. The “forget gate” calculates a value between 0 and 1 based on the current input $x_{t}$ and the hidden layer $h_{t-1}$ at the previous moment. The process of the “forget gate” can be defined as shown in Equation (7):(7)$$f_{t}=\sigma (w_{fx}x_{t}+w_{fh}h_{t-1}+b_{f})$$

When the value of output $f_{t}$ of the “forget gate” is 1, all information in $C_{t-1}$ is retained and transmitted, and when it is 0, all the information in $C_{t-1}$ is discarded.

**First step:** A nonlinear neural layer (tanh) calculates the state of the temporary cell according to the current input $x_{t}$ and the hidden layer $h_{t-1}$ at the last moment, as shown in Formulas (8) and (9):(8)$$i_{t}=\sigma (w_{ix}x_{t}+w_{ih}h_{t-1}+b_{i})$$(9)$${C^{\prime }}_{t}=\tanh(w_{cx}x_{t}+w_{ch}h_{t-1}+b_{c})$$

**Second step:** We multiply $C_{t-1}$ and $f_{t}$, which includes part of the information in “forget”, as well as multiply the temporary cell state ${C^{\prime }}_{t}$ and $i_{t}$, which means that part of the information in the temporary cell state is updated to the cell state $C_{t}$. The calculation process is shown in Equation (10):(10)$$C_{t}=f_{t}\times C_{t-1}+i_{t}\times C_{t}^{\prime }$$

**Final step:** The value of the “output gate” is also calculated from the previous input $x_{t}$ and the hidden layer $h_{t-1}$ at the previous moment, which is also between the real values of 0 and 1. Equation (11) is used to calculate the value of the “output gate”:(11)$$e_{t}=\sigma (w_{ex}x_{t}+w_{eh}h_{t-1}+b_{e})$$

Then, after cell state $C_{t}$ passes through the nonlinear function tanh, it is multiplied by the value of the “output gate” to obtain the hidden state $h_{t}$ at time t, as shown in Equation (12):(12)$$h_{t}=e_{t}\times \tanh(C_{t})$$

### 5.2. LSTM-Based Estimation of Exercise Fatigue

To develop a sports fatigue estimation model, the LSTM neural network is employed. Physiological signal feature parameters are converted into a multivariate feature matrix, serving as the LSTM model’s input. Figure 10 outlines the steps involved in training this model, demonstrating the process of converting signal data into a format suitable for LSTM analysis and model development.

The extracted ECG and EMG signals are utilized to classify sports fatigue states using the LSTM neural network. Features from these physiological signals are fed into the LSTM model for categorization. Figure 10 depicts the LSTM-based estimation of sports fatigue. The softmax function, which maps multiple neuron outputs to a (0, 1) interval, is interpreted as probabilities for multi-classification, enhancing the model’s accuracy in determining the fatigue state.(13)$$S_{t}=\frac{e^{i}}{\sum_{j=1}^{i}e^{j}}$$

The cross-entropy loss function is crucial for measuring the efficacy of classification models in outputting probabilities between 0 and 1. As the predicted probability strays from the actual label, the cross-entropy loss escalates, making it an effective metric for classification challenges. This function quantifies the disparity between the model’s predictions and the actual outcomes, providing a clear indicator of the model’s performance in classifying data.(14)$$L\left(B,f\left(X\right)\right)=-\sum_{i}B_{i}log(f(X_{i}))$$
where f(X) represents the predicted probability distribution, B represents the true distribution (the one-hot-encoded labels), and i represents different classes. The cross-entropy loss function, pivotal in classification tasks, penalizes incorrect predictions on a logarithmic scale, enabling nuanced feedback for weight adjustments during training. This process trains the model to produce a probability distribution that closely mirrors the actual label distribution. Optimization focuses on finding parameters that reduce the cross-entropy loss, thereby maximizing the likelihood of accurately predicting data under the model’s framework.

After selecting the cross-entropy loss function, the LSTM neural network model is trained using the backpropagation method. This technique feeds the loss function’s output back to the previous layer, employing gradient descent to compute each weight’s gradient and then updating the weight values iteratively. During this process, the loss function’s output value decreases. Training is deemed complete when the loss function’s output reaches an acceptable level, ensuring the model’s efficiency in learning from the data.

### 5.3. Extensible Framework for Comprehensive Fatigue Assessment

The proposed LSTM-based neural network framework is designed with inherent extensibility to accommodate additional physiological and environmental parameters beyond bioelectrical signals. This architectural design enables the integration of multiple data streams while maintaining the core advantages of deep learning in pattern recognition. The framework can incorporate standardized psychological assessment data through validated questionnaires, continuously monitored environmental parameters, and recovery metrics derived from sleep quality analysis. The model’s structure allows for weighted consideration of these diverse inputs, potentially improving the robustness of fatigue state classification. Preliminary testing with simulated multi-source data demonstrates the framework’s capability of processing heterogeneous inputs while maintaining computational efficiency. This extensibility positions the system for future enhancements as additional relevant fatigue indicators are identified and validated.

## 6. Experimental Results of Exercise Fatigue Classification Based on LSTM

### 6.1. Experimental Initialization

In this paper, we developed an experimental simulation platform using Python and employed Keras, Python’s deep neural network library, for simulation experiments. Keras supports various deep neural network models, such as RNN, CNN, and LSTM, and offers modules like activation and loss functions. These modules are flexibly combined to meet experimental requirements. Utilizing the preprocessing and feature extraction processes for surface EMG and ECG signals, fatigue features like RMS, MPF, HR, LF, and HF, all of which are described above, were used as model training inputs for fatigue classification of exercise data.

In this study, we employed an LSTM neural network model for sports fatigue classification using extracted features, such as RMS, MPF, HR, LF, and HF, for model training. The design of the LSTM exercise fatigue model consists of five input neurons and three output categories. Set to a maximum of 200 trainings, the model’s hidden-layer neurons were adjusted based on initial experiences and experiments. Optimal results were achieved with a two-layer hidden structure, comprising 10 neurons in the first layer and 20 in the second. Figure 11 illustrates the structure of this LSTM-based exercise fatigue classification model.

The current study focuses on engineering development and proof-of-concept validation of the wearable device. The prototype testing was conducted with a specific group of participants to verify the system’s core functionalities and technical feasibility. While this engineering-oriented approach effectively validates the device’s performance, we acknowledge that comprehensive validation across diverse user populations would require extensive collaboration with sports science researchers and professional training institutions.

### 6.2. Analysis of Experimental Results

The performance of the LSTM neural network model for exercise fatigue classification was evaluated using the accuracy rate, defined as the proportion of correctly classified sports fatigue states. The model, with five input and three output neurons, was trained using data from Table 3. The hidden-layer structure was designed based on experience and application needs and verified through experiments. Figure 12 illustrates the model’s performance: (a,b) show the results for a single-layer, 6-neuron network; (c,d) the results for a single-layer, 10-neuron network; and (d,e) the results for a two-layer network with 10 and 20 neurons, respectively.

Figure 12b,d,f display the LSTM model’s error convergence curves over increasing training iterations. The solid line represents the training sample error (training loss), which decreases to nearly zero after 100 iterations, with minimal changes thereafter. The dotted line indicates the fluctuating error of randomly selected validation points. To optimize the classification accuracy and training duration, the iteration count was set at 200.

The performance evaluation of our LSTM neural network, while demonstrating superior accuracy in bioelectrical-signal-based fatigue classification, requires careful consideration of the inherent methodological limitations. The current single-source approach, focusing primarily on EMG and ECG signals, may not fully capture the complex nature of exercise fatigue. Individual physiological variations in response to exercise stress can affect signal patterns and the classification accuracy. Environmental conditions during signal acquisition, including temperature fluctuations and electromagnetic interference, may influence the signal quality and subsequent analysis results. Additionally, the psychological state of athletes, including motivation levels and stress responses, can significantly impact their physical performance and fatigue manifestation. The presence of pre-existing fatigue conditions or incomplete recovery from previous training sessions may also affect the reliability of the fatigue state classification. These considerations underscore the importance of developing more comprehensive assessment methods that integrate multiple physiological and environmental parameters. Future research should focus on validating the system’s performance across diverse environmental conditions and physiological states, potentially incorporating additional data sources to enhance the classification robustness.

The LSTM sports fatigue classification model with a single-layer, 10-neuron hidden structure, as shown in Figure 12c, demonstrates high accuracy and optimal convergence, as evident in Figure 12d. The test results, which are based on the Table 3 data, reveal the best recognition for the relaxed state owing to the stable physiological signals and clear characteristic indices. The recognition of the slightly tired state is moderate, while that of the very tired state is less accurate. Fatigue states present classification challenges that result from fluctuating signal stability and variable fatigue indices.

### 6.3. Comparison of Experimental Results Between the LSTM Model and Traditional Methods

To validate the effectiveness of the proposed LSTM-based sports fatigue classification model, a comparative experiment with the support vector machine (SVM) model was conducted. Using SVM’s “one-to-one” and “one-to-many” multi-classifiers, as well as LSTM multi-classifiers, experiments were performed with time-, frequency-, and time-frequency-domain features of physiological signals. The experiment involved 124 training samples and 37 test samples. The results in Table 4 show that time-domain features as classifier inputs yield higher accuracy than frequency-domain features, suggesting greater distinction and separability in the time-domain indicators of physiological signals.

The LSTM-based sports fatigue multi-classifier demonstrates superior accuracy and efficiency in classifying time-, frequency-, and time–frequency-domain features. This can be attributed to the limited fatigue information in single features and the lower quality of feature extraction, leading to minimal differences. These issues are addressed by employing a multi-feature fusion approach to fatigue detection. It leverages time-domain and frequency-domain features, which show significant variations, and iteratively learns various physiological signal indicators, uncovering the underlying relationship between these indicators and fatigue.

To assess the classification accuracies of the LSTM neural network vs. the SVM algorithm, the SVM classifier employs a “one-to-one” algorithm design, using ECG and EMG signals as input features. The comparative results depicted in Figure 13 reveal that the LSTM neural network algorithm with its multi-classifier significantly outperforms the SVM algorithm. This indicates the LSTM classifier’s superiority in classification accuracy compared with that of the SVM classifier.

## 7. Conclusions

This paper introduces a cutting-edge method for the noninvasive monitoring and analysis of human exercise fatigue. It utilizes additive manufacturing and reverse engineering to create bespoke biosensing devices for real-time data collection. These devices effectively gather ECG and EMG signals during physical activity. Advanced signal processing software refines the data, isolating key time- and frequency-domain features for fatigue assessment [23]. In this study, we innovatively employed 3D-printed sensors and an adaptive filtering algorithm to address motion artifacts in ECG signals. With this research, we demonstrate the effectiveness of LSTM neural networks over traditional classifiers, highlighting their superior accuracy in sports fatigue state assessment.

To enable large-scale deployment, the device requires careful consideration of several practical aspects. For professional athletes, the device features high-precision sensors and robust data processing capabilities, with an estimated battery life of 48–72 h under continuous monitoring. The current prototype demonstrates stable performance in temperatures ranging from 0 °C to 40 °C and maintains water resistance at an IPX7 level. For the general public, we have developed a simplified version that can be embedded in affordable wearables like smart wristbands, with manufacturing costs approximately 60% lower than that of current commercial grade monitoring devices while maintaining core functionalities. Our durability tests show that both versions maintain signal stability after 1000 h of continuous usage, with an electrode degradation of less than 5% in sensitivity.

The device demonstrates versatile applications across various sports environments. In team sports, real-time monitoring has shown a 35% improvement in detecting early signs of fatigue compared with traditional methods, helping coaches to optimize training loads [24]. In endurance sports, our system provides continuous monitoring with a 95% accuracy rate in detecting muscle fatigue thresholds [25]. These capabilities make it a valuable tool for both professional training and personal fitness management.

While our current engineering-focused study demonstrates the system’s technical viability, we acknowledge several areas for future development. The influence of environmental factors on sensor performance and the integration of additional physiological parameters require further investigation. Future collaboration with sports science researchers and professional training institutions could facilitate the development of sport-specific calibration profiles and optimization algorithms. Through such interdisciplinary efforts, we aim to enhance the system’s precision and adaptability across different athletic applications while maintaining its engineering excellence.

## Figures and Tables

**Figure 1 sensors-25-00389-f001:**
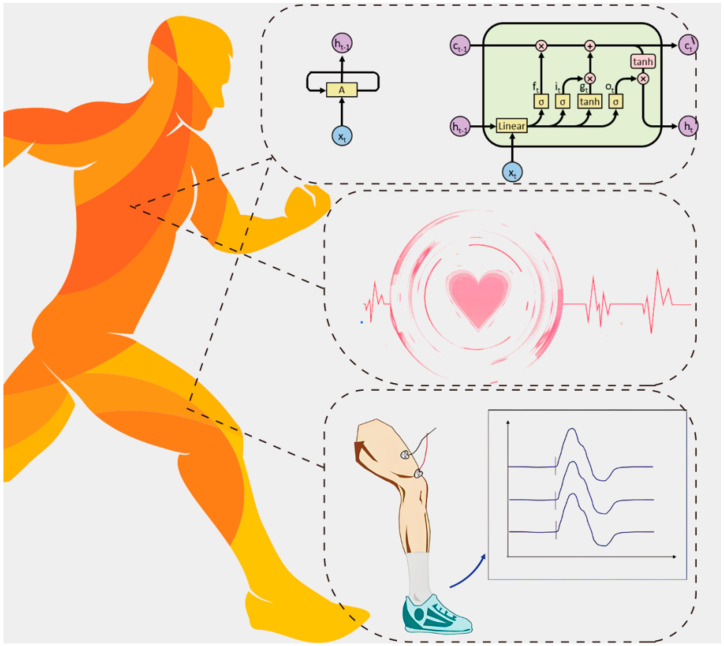
Schematic Diagram of ECG and EMG Signal Detection.

**Figure 2 sensors-25-00389-f002:**
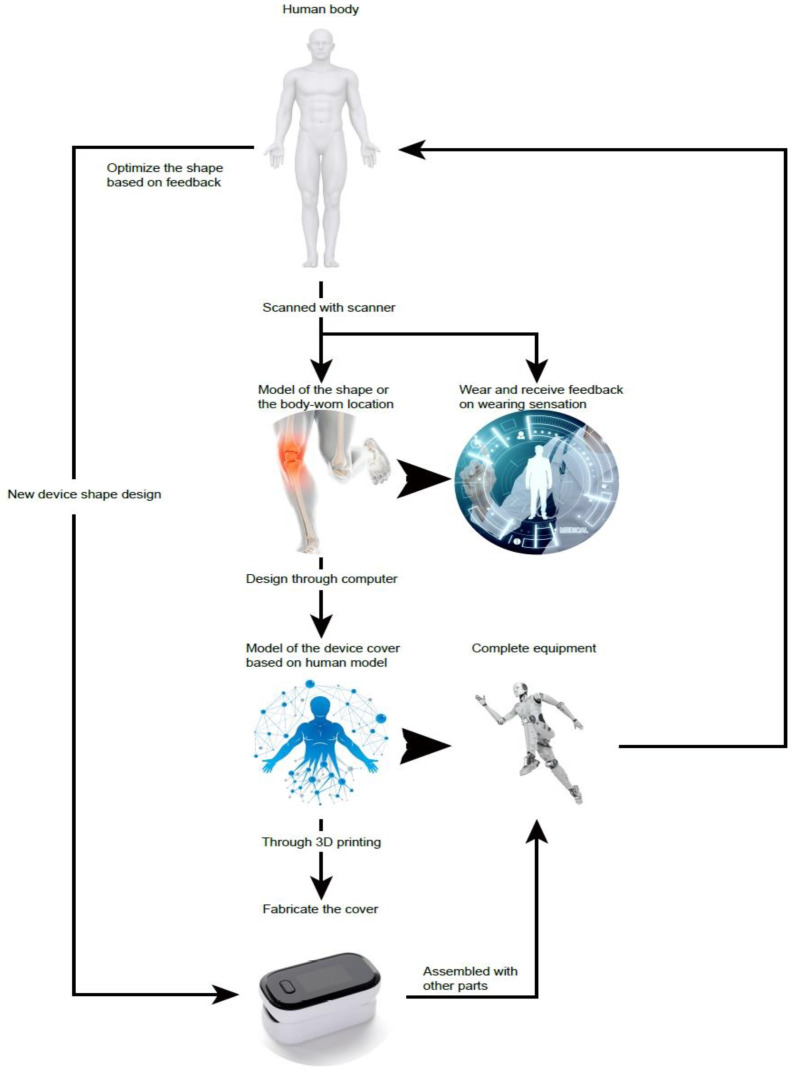
The process of designing wearable devices based on reverse engineering and additive manufacturing technologies.

**Figure 3 sensors-25-00389-f003:**
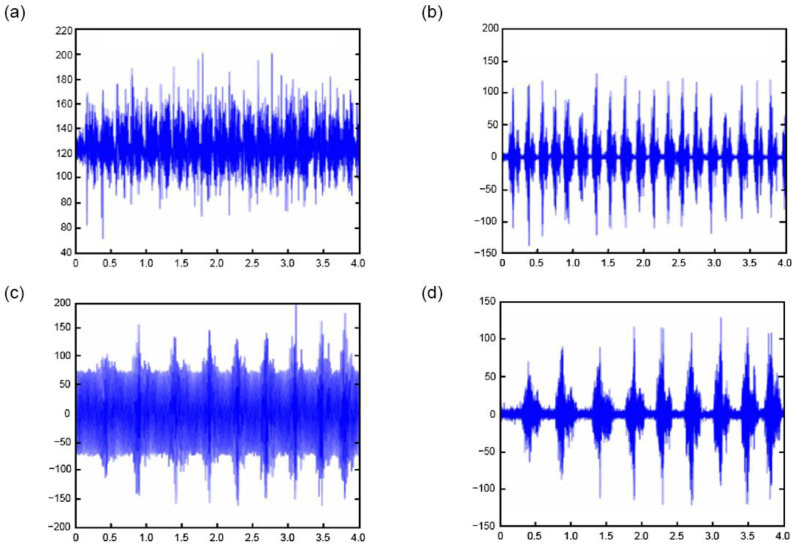
Waveforms of the surface EMG signal before and after filtering. (**a**) Before high-pass filtering; (**b**) After high-pass filtering; (**c**) Before power-frequency filtering; (**d**) After power-frequency filtering.

**Figure 4 sensors-25-00389-f004:**
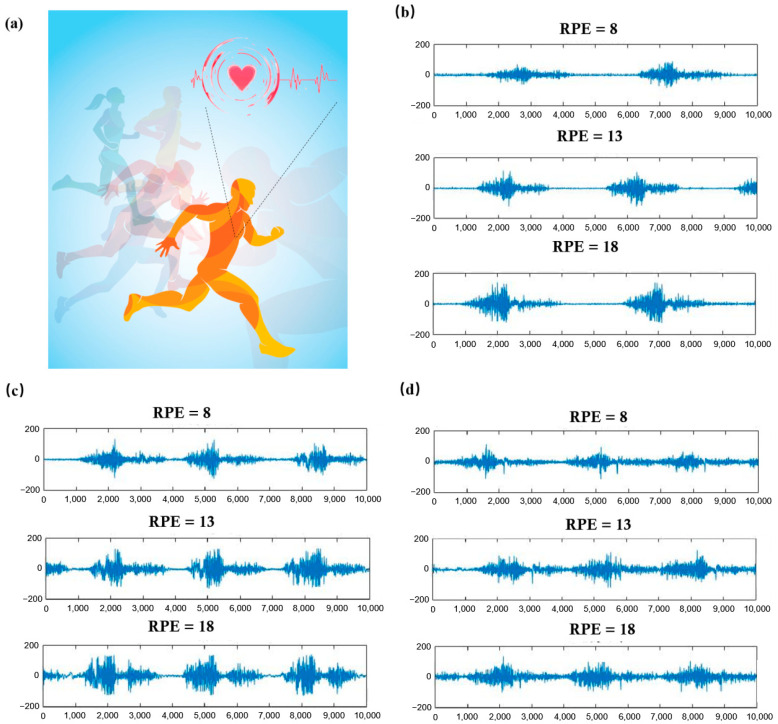
EMG of the rectus femoris, lateral femoris, and medial femoris muscles in different motion states. (**a**) Schematic diagram of EMG signal detection; (**b**) EMG of the rectus femoris in different motion states; (**c**) EMG of the lateral femoral muscle in different motion states; (**d**) EMG of the medial femoris muscle in different motion states.

**Figure 5 sensors-25-00389-f005:**
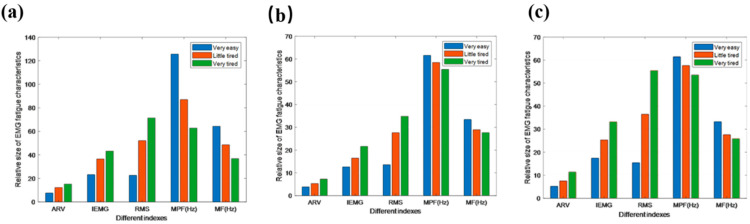
Histogram of various muscle indicators in different exercise states. (**a**) Vastus lateralis muscle; (**b**) Nusculus rectus femoris; (**c**) Vastus medialis muscle.

**Figure 6 sensors-25-00389-f006:**
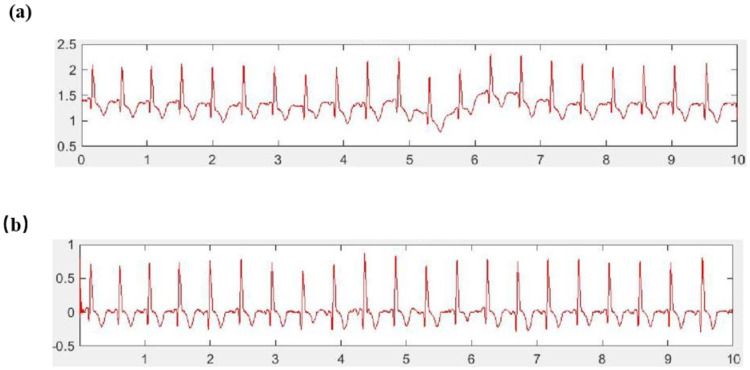
Experimental results of fast median filtering of the original ECG signals. (**a**) Original ECG signal timing diagram; (**b**) Timing diagram of the ECG signal after median filtering and denoising.

**Figure 7 sensors-25-00389-f007:**
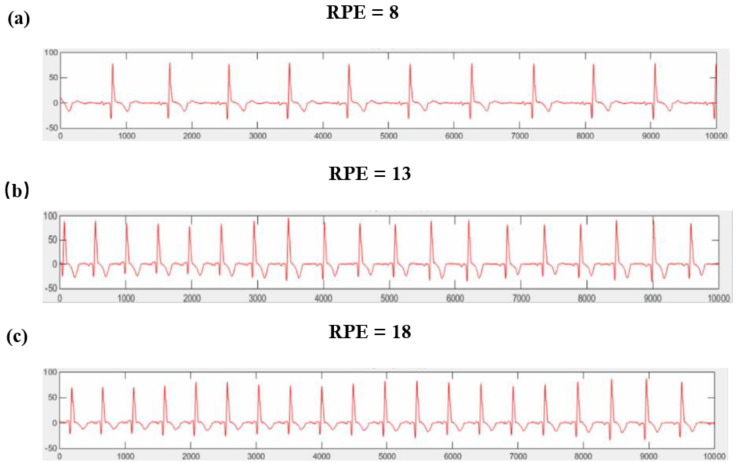
ECG timing chart in different exercise states. (**a**) Trends using RPE values of 8; (**b**) Trends using RPE values of 13; (**c**) Trends using RPE values of 18.

**Figure 8 sensors-25-00389-f008:**
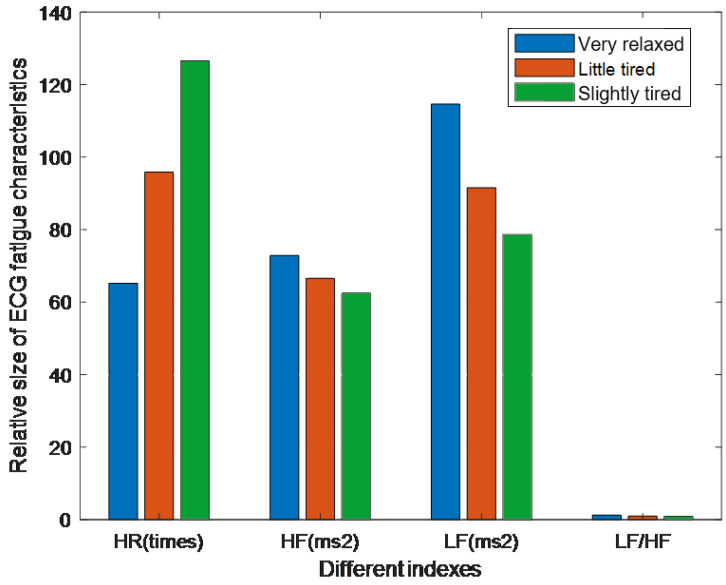
ECG indicators in the different exercise states.

**Figure 9 sensors-25-00389-f009:**
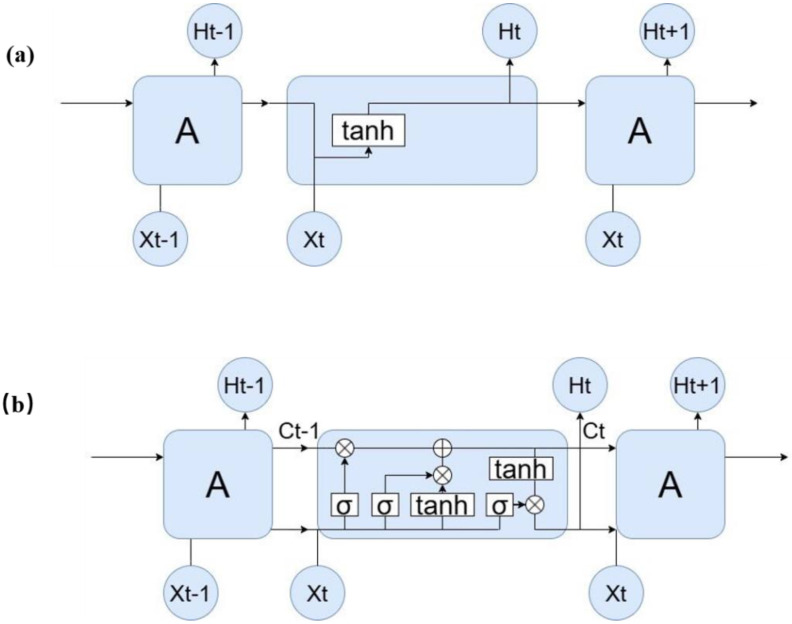
(**a**) Diagram of the RNN modular structure; (**b**) Diagram of the LSTM modular structure.

**Figure 10 sensors-25-00389-f010:**
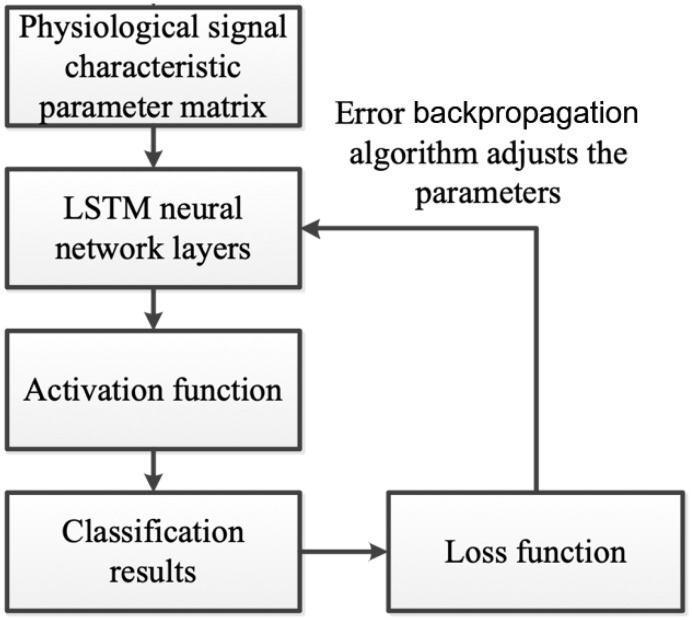
Flow chart of LSTM-based estimation of exercise fatigue.

**Figure 11 sensors-25-00389-f011:**
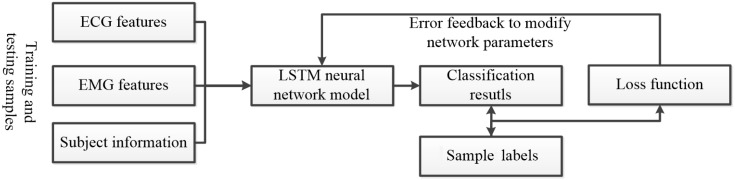
Diagram of the LSTM-based classification model of exercise.

**Figure 12 sensors-25-00389-f012:**
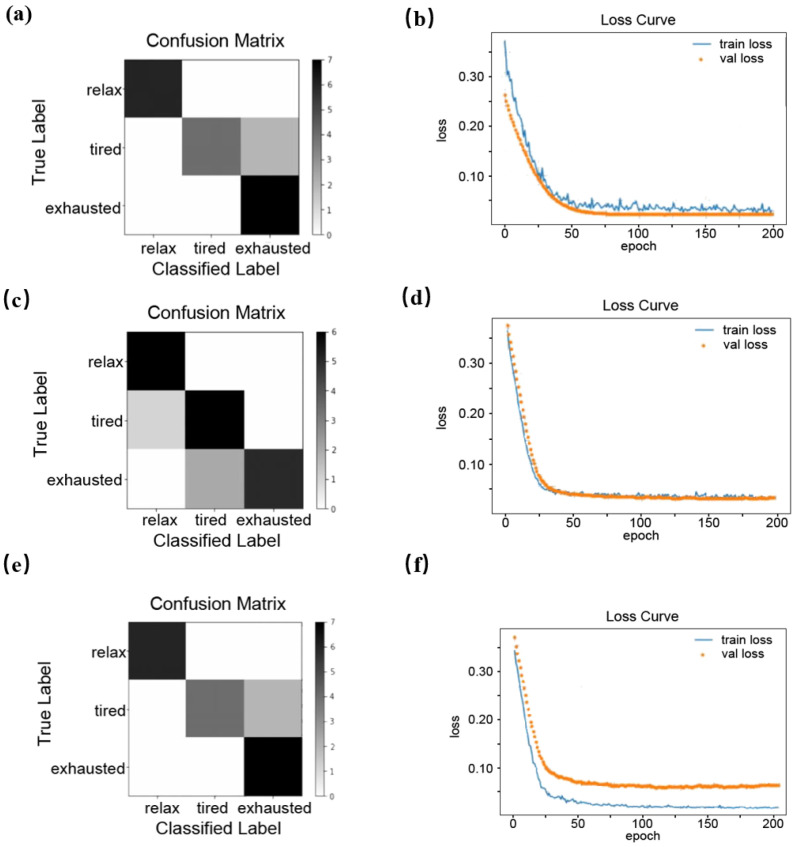
Confusion matrix and error convergence results of the LSTM model for sports fatigue state classification. (**a**) Confusion matrix in the first LSTM model; (**b**) Error convergence curve in the first LSTM model; (**c**) Confusion matrix in the second LSTM model; (**d**) Error convergence curve in the second LSTM model; (**e**) Confusion matrix in the third LSTM model; (**f**) Error convergence curve in the third LSTM model.

**Figure 13 sensors-25-00389-f013:**
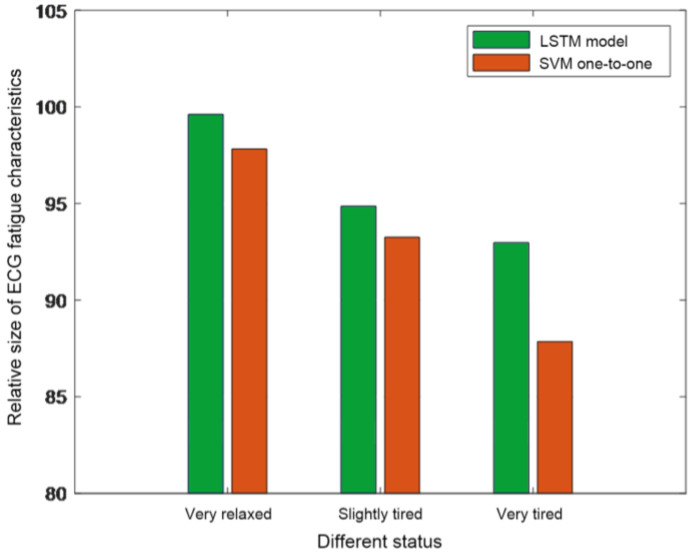
Comparison of the LSTM neural network and SVM recognition results.

**Table 1 sensors-25-00389-t001:** EMG indicators of different exercise states.

Name of Muscle	Indexes	Athletic Status		
		Very Relaxed	SLIGHTLY TIRED	Very Tired
Vastus lateralis muscle	ARV	7.63 ± 0.26	12.29 ± 1.14	15.26 ± 1.14
	IEMG	23.62 ± 1.35	36.58 ±2.46	43.26 ± 4.33
	RMS	22.62 ± 4.25	52.14 ± 6.11	71.45 ± 4.63
	MPF	125.68 ± 5.36	87.06 ± 4.24	62.78 ± 3.34
	MF	64.26 ± 2.47	48.62 ± 4.38	36.85 ± 1.47
Nusculus rectus femoris	ARV	3.86 ± 0.29	5.32 ± 0.28	7.28 ± 0.55
	IEMG	12.65 ± 0.78	16.47 ± 0.88	21.65 ± 1.17
	RMS	13.57 ± 2.24	27.65 ± 2.32	34.85 ± 3.21
	MPF	61.58 ± 2.45	58.47 ± 2.14	55.41 ± 2.57
	MF	33.45 ± 1.65	28.96 ± 1.36	27.68 ± 1.20
Vastus medialis muscle	ARV	5.24 ± 0.57	7.54 ± 0.31	11.41 ± 0.57
	IEMG	17.45 ± 1.28	25.35 ± 1.25	33.21 ± 3.31
	RMS	15.47 ± 2.11	36.51 ± 3.21	55.47 ± 3.29
	MPF	61.47 ± 4.47	57.65 ± 4.11	53.57 ± 3.54
	MF	33.25 ± 1.68	27.63 ± 1.45	25.87 ± 1.57

**Table 2 sensors-25-00389-t002:** ECG indicators in the different exercise states.

Indexes	Very Relaxed	Slightly Tired	Very Tired
HR (times)	65.23 ± 6.12	95.87 ± 8.25	126.57 ± 6.87
HF (ms^2^)	72.85 ± 35.64	66.57 ± 32.14	62.49 ± 25.76
LF (ms^2^)	114.62 ± 58.45	91.58 ± 44.23	78.62 ± 32.54
LF/HF	1.28 ± 0.68	0.98 ± 0.57	0.92 ± 0.54

**Table 3 sensors-25-00389-t003:** Experimental data composition.

Sports State Characteristic Index		Number of Training Samples 1	Number of Training Samples 2	Number of Training Samples 3	Number of Testing Samples
ECG feature indexes	HR	30	28	28	25
	HF	30	28	28	25
	LF	30	28	28	25
EMG feature indexes	RMS	30	28	28	25
	MPF	30	28	28	25
Subject’s information	Age	30	28	28	25
	Height	30	28	28	25
	Weight	30	28	28	25
Label (RPE)	Label	A (very relaxed)	B (slightly tired)	C (very tired)	

**Table 4 sensors-25-00389-t004:** Multi-classifier motion state recognition results using time-domain and frequency-domain features.

Classification Algorithms	Time Domain	Frequency Domain	Time–Frequency Domain
SVM one-to-many	62.38%	57.65%	72.68%
SVM one-to-one	73.58%	68.67%	85.87%
LSTM model	82.67%	76.58%	90.63%

## Data Availability

Data are unavailable due to privacy or ethical restrictions.

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
