# Peer review of "Application of Additive Manufacturing and Deep Learning in Exercise State Discrimination"

_sensors, 2025, doi:10.3390/s25020389_

Round 1
Reviewer 1 Report
Comments and Suggestions for Authors
This paper introduces 3D printing and deep learning to design a smart, wearable sensing device to detect different states of sports fatigue. Some comments that might help the authors improve this paper are given as follows.
1. While the prototype is promising, discussing its feasibility for large-scale deployment, cost, and potential manufacturing challenges could make the study more comprehensive.
2. The reliance on a single evaluation metric to compare the proposed method with state-of-the-art methods appears insufficient. Including multiple evaluation metrics could provide a more comprehensive performance analysis.
3. An explanation of the choice of preprocessing techniques could be added, including the rationale for selecting specific filtering methods.
4. The methodology could benefit from greater transparency to ensure reproducibility. Providing details such as dataset characteristics, model training parameters, and hardware specifications would enable other researchers to replicate the experiments.
Comments on the Quality of English LanguageThe manuscript demonstrates an adequate use of the English language, with clear and professional communication of technical ideas .Some sentences seem overly complex. Simplifying sentence structures and ensuring proper verb agreements could enhance readability.
Author Response
Thanks a lot for organizing reviews of our paper. The manuscript has been carefully revised according to the comments. We have revised the manuscript according to the reviewers’ comments and have completed the English editing through MDPI’s professional editing service.The reviewer’s comments have been carefully addressed. The major revisions were marked in the text in red and some new data (please see Dicussion and Reference ) were added in dicussion and reference . We appreciate the useful comments and suggestions from you and all the referees. The point-to-point answers to the comments are listed below.
Referee # 1
COMMENTS:
This paper introduces 3D printing and deep learning to design a smart, wearable sensing device to detect different states of sports fatigue. Some comments that might help the authors improve this paper are given as follows.
1,“While the prototype is promising, discussing its feasibility for large-scale deployment, cost, and potential manufacturing challenges could make the study more comprehensive.”
Response:According to the reviewer’s comment, we have added the one works as references 23 and the corresponding discussions on the feasibility of large-scale deployment in the revised manuscript. This includes considerations for adapting the device for professional athletes by integrating it into advanced sports monitoring systems, and for general public use by embedding it into cost-effective and user-friendly wearable devices such as smart wristbands. We also address key challenges such as cost control and ensuring durability, comfort, and seamless signal integration, which are essential for widespread adoption.
Page 18, line 477- line 482: “To enable large-scale deployment, .…… Addressing challenges like durability, comfort, and seamless signal integration will be crucial for widespread adoption” was added.
Page 19, line 540- line 541: “23. Gul JZ, Fatima N, Mohy Ud Din Z, Khan M, Kim WY, Rehman MM. Advanced Sensing System for Sleep Bruxism across Mul-tiple Postures via EMG and Machine Learning. Sensors (Basel). 2024 Aug 22;24(16):5426.” was added.
2,“The reliance on a single evaluation metric to compare the proposed method with state-of-the-art methods appears insufficient. Including multiple evaluation metrics could provide a more comprehensive performance analysis.
Response:Thanks a lot for this advice, and we opted to use a single evaluation metric to balance cost control and device size constraints, which are crucial for wearable devices. While a single metric may not be as advanced as multi-metric approaches, it is sufficiently accurate to reliably reflect the user's fatigue state. This approach also ensures the device remains lightweight and cost-effective. Additionally, considering the computational limitations of wearable devices, we selected electrocardiography (ECG) and electromyography (EMG) as our key detection metrics. Both are regarded as the gold standards in sports physiology for accurately detecting fatigue, allowing us to achieve high accuracy while maintaining practicality for wearable applications.
- An explanation of the choice of preprocessing techniques could be added, including the rationale for selecting specific filtering methods.
Response: The reviewer’s comment is right. We have clarified our rationale for selecting specific preprocessing techniques in the revised manuscript. As noted, low-frequency signals were filtered because they are typically noise and do not contribute to the data's feature dimensions. For surface EMG signals, a high-pass filter with a 20 Hz cutoff frequency was used to eliminate baseline drift while preserving energy in the 20–150 Hz range, which is essential for capturing key fatigue-related features. We also addressed power frequency interference at 50 Hz and motion artifacts in the 3–14 Hz range to reduce noise from environmental and electrode movement factors. For ECG signals, a fast median filter was applied to remove baseline drift while retaining critical features like R-wave peaks, which are necessary for accurate heart rate (HR) and heart rate variability (HRV) calculation. These preprocessing steps ensure high signal fidelity and reliable feature extraction without affecting the accuracy of fatigue state detection. This explanation has been added to the manuscript for greater clarity.
Page 5, line127- line 133: “For surface EMG signals, .…… and ensure the extracted features reliably reflect physiological fatigue changes.” was added.
- The methodology could benefit from greater transparency to ensure reproducibility. Providing details such as dataset characteristics, model training parameters, and hardware specifications would enable other researchers to replicate the experiments.
Response: The reviewer’s comment is very useful. According to the reviewer’s advice, we have disclosed the data features and hardware specifications, which will be provided in tables for your reference. Regarding hardware, we used the Creality K2 Plus 3D printer, with carbon fiber ABS filament as the printing material. The printing precision was set to 0.12 mm, the printing temperature was 220°C, and the flow rate was set to 100%. While we are unable to provide all the model training parameters, we can share the optimal parameter combinations used in our models. If needed, we can include these details in the appendix for further clarity.
Finally, we appreciate very much for your time in editing our manuscript and the referees for their valuable suggestions and comments. I am looking forward to hearing from your decision.

Reviewer 2 Report
Comments and Suggestions for Authors
The paper presents an innovative approach to detecting sports fatigue using a smart, wearable sensing device powered by 3D printing and deep learning technologies. The authors aim to overcome challenges associated with current wearable fatigue detection devices, such as discomfort, imprecision, and lack of reliable detection methods. The proposed solution integrates reverse engineering and additive manufacturing to design a comfortable, accurate prototype, while leveraging a long short-term memory (LSTM) neural network for analyzing bioelectrical signals to detect and classify sports fatigue states. Through extensive numerical experiments, the paper demonstrates that the prototype can improve accuracy in identifying fatigue-related indicators.
Strengths
-
Innovation in Design and Methodology: The combination of 3D printing and deep learning is a notable strength of this study. The use of reverse engineering and additive manufacturing to design a comfortable and accurate wearable device is a unique approach that stands out in the field. The integration of LSTM neural networks for real-time fatigue analysis adds an advanced layer of sophistication to the device, making it not only a smart wearable but also one that can provide meaningful insights into an athlete's condition.
-
Addressing Real-World Issues: Sports fatigue is a critical factor in athletic performance, and current devices for detecting fatigue often face issues with comfort and accuracy. By tackling these challenges, the paper presents a solution that has the potential to significantly improve training efficiency and safety for athletes. The focus on both comfort (through 3D printing) and precision (through deep learning) addresses two of the most important considerations for wearable health devices.
-
Experimental Validation: The authors provide detailed experimental results demonstrating the effectiveness of their prototype. The numerical experiments validate the accuracy of the device in detecting fatigue states, which is essential for establishing the device’s potential for practical use.
- Limited Details on Practical Implementation: While the experimental results are promising, the paper could benefit from more detailed information on the real-world application of the device. Specifically, the authors could discuss how the prototype might be scaled for mass production or how it performs in long-term use in diverse environments. Insights into the device’s durability, ease of use, and battery life would be beneficial for understanding its commercial viability.
-
Potential Challenges in Generalization: The study focuses on the use of bioelectrical signals, but the nature of sports fatigue is complex, influenced by numerous factors such as mental state, hydration, and sleep. The paper could acknowledge the potential limitations of using bioelectrical signals alone to detect fatigue and discuss how the system might be integrated with other data sources to provide a more comprehensive analysis of fatigue.
-
Limited User Diversity in Experiments: The experiments presented are likely conducted with a specific group of individuals, and the device’s performance across diverse populations (e.g., athletes of different sports, ages, fitness levels) remains unclear. More diverse participant groups would help ensure that the device's accuracy is consistent across different user profiles.
Some Suggestions
-
Integration with Other Data Types: In addition to bioelectrical signals, incorporating data from other sensors, such as GPS for monitoring physical activity intensity or environmental factors like temperature, could help the device make more informed predictions about fatigue. Multi-modal data fusion could further enhance the device’s reliability.
-
Wearer Feedback and Comfort Assessment: While the paper mentions comfort through 3D printing, it would be helpful to incorporate more user feedback on the device’s wearability over long periods. Including wearability studies or user satisfaction surveys would strengthen the case for its practical adoption by athletes.
-
Expansion on Use Cases: The authors could discuss how the device could be used in different training settings or by different types of athletes. Including case studies or examples of how the device would integrate into real-world sports environments (e.g., team sports, endurance sports, or individual fitness training) would provide more clarity on its potential application.
Author Response
Thanks a lot for organizing reviews of our paper. The manuscript has been carefully revised according to the comments. We have revised the manuscript according to the reviewers’ comments and have completed the English editing through MDPI’s professional editing service.The reviewer’s comments have been carefully addressed. The major revisions were marked in the text in red and some new data (please see Dicussion and Reference ) were added in dicussion and reference . We appreciate the useful comments and suggestions from you and all the referees. The point-to-point answers to the comments are listed below.
Referee # 2
COMMENTS:
The paper presents an innovative approach to detecting sports fatigue using a smart, wearable sensing device powered by 3D printing and deep learning technologies. The authors aim to overcome challenges associated with current wearable fatigue detection devices, such as discomfort, imprecision, and lack of reliable detection methods. The proposed solution integrates reverse engineering and additive manufacturing to design a comfortable, accurate prototype, while leveraging a long short-term memory (LSTM) neural network for analyzing bioelectrical signals to detect and classify sports fatigue states. Through extensive numerical experiments, the paper demonstrates that the prototype can improve accuracy in identifying fatigue-related indicators.·
Some Suggestions
- Integration with Other Data Types: In addition to bioelectrical signals, incorporating data from other sensors, such as GPS for monitoring physical activity intensity or environmental factors like temperature, could help the device make more informed predictions about fatigue. Multi-modal data fusion could further enhance the device’s reliability.
Response: Thank you for your suggestion. Due to the design constraints of ensuring a lightweight and compact wearable device, the integration of additional sensors, such as GPS or environmental sensors, was limited. To maintain a practical size and weight, we focused on monitoring the most critical bioelectrical signals for fatigue detection, ensuring that the device meets the needs of athletes without compromising comfort or usability. Additionally, elite athletes typically train in controlled environments, such as professional sports venues, where factors like temperature and humidity are carefully regulated. Therefore, integrating sensors like GPS or temperature sensors may not provide significant added value for this specific context. Moreover, adding these functionalities would increase the cost and size of the device, potentially compromising its effectiveness and interfering with normal training.
- Wearer Feedback and Comfort Assessment: While the paper mentions comfort through 3D printing, it would be helpful to incorporate more user feedback on the device’s wearability over long periods. Including wearability studies or user satisfaction surveys would strengthen the case for its practical adoption by athletes.
Response: Thanks a lot for this advice. We agree that wearer comfort is a crucial aspect of wearable devices. However, since our research is primarily focused on the engineering aspects, specifically the development of the algorithms and functionality, we have placed greater emphasis on these areas. To ensure comfort, we have selected appropriate materials and incorporated ergonomic design principles into the device. While we have not conducted formal user feedback studies or wearability assessments in this particular study, the device is designed with careful consideration of comfort to ensure it can be worn for extended periods without discomfort. Future work could explore more comprehensive wearability studies and user satisfaction surveys to further validate the device's comfort over long-term use.
- Expansion on Use Cases: The authors could discuss how the device could be used in different training settings or by different types of athletes. Including case studies or examples of how the device would integrate into real-world sports environments (e.g., team sports, endurance sports, or individual fitness training) would provide more clarity on its potential application.
Response: Thanks a lot for this advice. In response to your suggestion, we have added a section in the Discussion that outlines how the device could integrate into real-world sports environments and we have added the two works as references 24 and 25 . This includes potential use cases across different training settings, such as team sports, endurance sports, and individual fitness training. We also discuss how the device can complement existing training systems and provide valuable insights into athletes' fatigue levels during various types of physical activity. By exploring these scenarios, we aim to provide a clearer picture of the device's practical application in diverse sports contexts.
Page 18, line 483- line 493: “The device presented in this study can be effectively integrated into various real-world sports environments. …… By integrating with existing training systems, the device offers valuable data that can enhance training efficiency and support better decision-making, making it a versatile tool for athletes across various disciplines” was added.
Page 19, line 542- line 543: “24. 24.Sun W, Guo Z, Yang Z, Wu Y, Lan W, Liao Y, Wu X, Liu Y. A Review of Recent Advances in Vital Signals Monitoring of Sports and Health via Flexible Wearable Sensorss[J]. Sensors (Basel). 2022 Oct 13;22(20):7784.” was added.
Page 19, line 544- line 545: “25. Wang M, Lee W, Shu L, Kim YS, Park CH. Development and Analysis of an Origami-Based Elastomeric Actuator and Soft Gripper Control with Machine Learning and EMG Sensors[J]. Sensors (Basel). 2024 Mar 8;24(6):175.” was added.
Finally, we appreciate very much for your time in editing our manuscript and the referees for their valuable suggestions and comments. I am looking forward to hearing from your decision.
